# The Push-Out Bond Strength, Surface Roughness, and Antimicrobial Properties of Endodontic Bioceramic Sealers Supplemented with Silver Nanoparticles

**DOI:** 10.3390/molecules29184422

**Published:** 2024-09-18

**Authors:** Karla Navarrete-Olvera, Nereyda Niño-Martínez, Idania De Alba-Montero, Nuria Patiño-Marín, Facundo Ruiz, Horacio Bach, Gabriel-Alejandro Martínez-Castañón

**Affiliations:** 1Doctorado Institucional en Ingeniería y Ciencia de Materiales, Universidad Autónoma de San Luis Potosi, Sierra Leona No. 550 Col. Lomas 2da. Sección, San Luis Potosí 78210, Mexico; karla.navarreteolvera@gmail.com; 2Facultad de Ciencias, Universidad Autónoma de San Luis Potosi, Av. Parque Chapultepec No. 1570, Privadas del Pedregal, San Luis Potosí 78295, Mexico; nereyda.nino@uaslp.mx (N.N.-M.); idania.dealba@uaslp.mx (I.D.A.-M.); facundo@fciencias.uaslp.mx (F.R.); 3Facultad de Estomatología, Universidad Autónoma de San Luis Potosi, Av. Dr. Manuel Nava No. 2, Zona Universitaria, San Luis Potosí 78290, Mexico; nuriapaty@uaslp.mx; 4Faculty of Medicine, Division of Infectious Diseases, University of British Columbia, Vancouver, BC V6H 3Z6, Canada; hbach@mail.ubc.ca

**Keywords:** antibacterial activity, bioceramic sealers, physiochemical properties, push-out bond strength, root canal filling, silver nanoparticles

## Abstract

This study evaluated push-out bond test (POBT), surface roughness, and antimicrobial properties against *Enterococcus faecalis* of bioceramic sealers supplemented with silver nanoparticles (AgNPs). The sealers tested were CeraSeal^®^, EndoSequence^®^ BC Sealer^TM^, and Bio-C^®^ Sealer. The POBT was measured with a Universal Testing Machine, and the type of failure was evaluated with a stereomicroscope. The roughness average (Sa) and peak–valley height (Sy) values were evaluated by atomic force microscopy. The bacterial growth inhibition was evaluated using a disk diffusion test, and antimicrobial activity was determined with the plate microdilution method. The POBT showed no significant difference between sealers with and those without NPs in cervical and apical thirds (*p* > 0.05). In the middle third, the adhesion force was significant for Endosequence BC Sealer^®^ (*p* < 0.05). The results showed that the Sa and Sy parameters, when AgNPs were added, did not show a statistically significant difference compared to the groups without nanoparticles (*p* > 0.05). All tested sealers showed bacterial growth inhibition, but no significant difference was found. Their efficacy, in descending order of antibacterial activity when AgNPs were added, is as follows: EndoSequence^®^ BC Sealer^TM^ > Bio-C^®^ Sealer > CeraSeal^®^. The incorporation of AgNPs into bioceramics improves antimicrobial activity without affecting mechanical properties.

## 1. Introduction

The ultimate biological goal of root canal treatment is to prevent and repair apical periodontitis (AP); complete resolution of AP after initial treatment or retreatment ranges between 74% and 86% and functionality over time in 91–97% of cases [1,2]. The etiology of AP is well established, and the microorganisms involved are the main causal factor [3]. The filling of the root canal has been described as the most critical component of treatment, to seal and isolate the root canal from irritants that remain after shaping and cleaning [4,5]. The objectives of the obturation are to prevent the entry of oral fluids, trap residual bacteria and deprive them of nutrients, and provide an apical seal [6,7].

Various filling materials have been used, including a combination of core materials such as gutta-percha. However, it presents a lack of adherence to the dentin, and due to this limitation, a sealer is necessary [8,9]. Sealer are thin, sticky pastes that function as a lubricant and fixing agent during obturation and can fill lateral and accessory canals [10,11].

An ideal sealer should provide characteristics like easy entry to the canal, sealing the canal laterally and apically; it should not shrink after being inserted; be bacteriostatic and radiopaque; not stain the dental structure; make a hermetic seal; be sticky and adhesive to the dentin and the filling material; be dimensionally stable; have longer working time; be insoluble in tissue fluids; be biocompatible, including being non-mutagenic, non-sensitive, and non-cytotoxic; be capable of being removed for retreatment; and be bioactive [10,12,13].

Since the introduction of mineral trioxide aggregate (MTA), the sealers based on calcium silicate or bioceramics have been developed for clinical procedures, including endodontic sealers. New formulations have been proposed to improve problems such as working time, cost, and handling difficulties; their pH is above 12, they have antibacterial properties and biocompatibility, and they expedite the regeneration of periapical tissue [14,15,16].

Bioceramics are metallic and non-metallic, inorganic, biocompatible materials that have been extensively used in medical sciences for the replacement of joints, bone tissues, and heart valves as well as cochlear replacement. They are chemically stable, non-corrosive, and interact well with organic tissue [17]. Bioceramics are classified as (1) bioinert, which produces an insignificant response in the stimulating tissue without having any biological effect, including alumina and zirconia; (2) bioactive, interacting with the surrounding tissue to promote the growth of tissues, like calcium silicate, bioactive glass, bioactive glass ceramic, and hydroxyapatite; and (3) biodegradable, finally replacing or are being incorporated into the tissues, like calcium phosphate and bioactive glass. In endodontics, the bioceramics used are generally bioactive, including calcium silicate-based cements, due to their excellent physical and chemical properties; among them are biocompatibility and bioactivity [18,19].

Modifications have been made to root canal sealers to improve their properties; these efforts have focused mainly on incorporating antibacterial additives such as silver [7].

Nanomaterials show promise in antibacterial therapies due to their physicochemical properties, such as nanosizes (10^−9^ m), large surface-to-mass ratio, and increased chemical reactivity [20]. Their antimicrobial effects are superior to their bulk counterparts due to higher surface area and the charge density resulting in greater interaction with microbial cells [21].

Because of increasing bacterial resistance, research into the antibacterial activity of silver nanoparticles (AgNPs) has increased. It has been shown that at low concentrations (1 wt%), Ag^+^ is not toxic to human cells [22,23]. Thus, AgNPs have been incorporated into biomaterials to prevent or reduce biofilm formation. Due to the higher surface/volume ratio and the small size of the particles, they have excellent antimicrobial action without affecting the mechanical properties of the materials [24]. Furthermore, silver nanoparticles (AgNPs) of 1–10 nm adhere to the cell membrane surface and interfere with bacterial metabolism, permeability, and respiration [25]. The Ag^+^ delivered by NPs to the bacteria promotes lysis [26].

Adding NPs in sealers has been primarily proposed to increase antibacterial activity, substantivity, and diffusion within dentinal tubules [27,28]. The addition of AgNPs promotes the antibacterial activity of Ca (OH)_2_ and other filling materials like silicate cement, such as MTA and Portland cement [20].

*Enterococcus faecalis* is a gram-positive facultative anaerobe, which has been reported by various studies as a highly resistant microorganism and associated with persistent infections in endodontics with high prevalence values reaching up to 90% [29]. The ability of *E. faecalis* to form biofilms in root canals and function as a monoinfection in treated canals without synergistic support from other bacteria makes it considered a highly resistant pathogen to antimicrobial agents and a standard for testing antimicrobial materials and agents in endodontics [30].

Considering the relevance of bioceramic materials in contemporary endodontics, it is important to improve their properties without affecting their already reported advantages. Therefore, the aim of study was to determine the effect of AgNps supplementation on bioceramic sealers’ physical-mechanical properties and antimicrobial activity. The null hypothesis on this research was that adding AgNPs to bioceramic sealers does not modify their physical-mechanical properties and improves their antimicrobial activity.

## 2. Results

### 2.1. Synthesis of AgNPs

Analysis of the synthesized AgNPs (Figure 1) showed a spherical morphology and a diameter of 5.57 nm. The hydrodynamic diameter obtained from DLS was 5.6 nm with a surface plasmon resonance maximum at 405 nm confirming their identity. A zeta potential of −36 mV was found with a polydispersity index of 24.36%.

### 2.2. Push-Out Bond Strength Test

The test showed no significant difference between the sealers with or without AgNPs in the cervical third (*p* > 0.05) (Table 1). In the middle third, no statistically significant differences were found between G1 and G2 nor between G5 and G6 (*p* > 0.05); however, a statistically significant decrease in adhesion strength was observed between G3 and G4 (*p* < 0.05).

Lastly, the results in the apical third showed no significant difference between endodontic sealers with and those without AgNPs (*p* > 0.05) (Table 1).

### 2.3. Type of Failure

The frequency and percentage of the type of failure in the cervical third are presented in Figure 2A. For the groups without AgNPs, G1, G3, and G5, the most prevalent type of failure was mixed, with 75%, 55%, and 80%, respectively. For groups with AgNPs, G2 and G6, the most frequent was the cohesive failure, with 55% for both, whereas in G4, the mixed failure was the most prevalent with 65%.

The type of failure in the medium third is presented in Figure 2B. For the groups without AgNPs, G1 and G5, the most prevalent was a mixed failure, with 70% and 55%, respectively. For G3, the most frequent was the cohesive failure, with 75%. For groups with AgNPs, G2 and G6, the most frequent was a cohesive failure, with 85% for both, whereas in G4, the mixed failure was the most prevalent, with 65%. Results in the apical third are presented in Figure 2C. For all groups, the most frequent was a cohesive failure.

### 2.4. Surface Roughness Measurement

The mean surface roughness values of the sealers tested, and the 3D atomic force microscopy images, are presented in Table 2 and Figure 3. The statistical analysis showed that both the Sa and Sy parameters in the group with AgNPs were not statistically significant compared with the groups without AgNPs (*p* > 0.05). The lowest roughness values (Sa and Sy) in the groups with AgNPs were observed in the CeraSeal^®^ and the highest surface roughness value was found in the EndoSequence^®^ BC Sealer^TM^ group without significant differences.

### 2.5. Disk Diffusion Test

The inhibition halos for each study group showed no statistical differences between endodontic sealers with and those without AgNPs (Table 3).

### 2.6. Plate Microdilution Method

The addition of AgNPs into bioceramic sealers improved bacterial growth inhibition in Endosequence BC Sealer™ and Bio-C Sealer (*p* < 0.05), but not in CeraSeal^®^ (Table 4).

## 3. Discussion

Despite the efforts, no protocol for cleaning and disinfection allows the eradication of total bacteria from the root canal system [31]. Endodontic sealers help minimize leakage, provide antimicrobial activity by reducing the potential for residual bacteria, and resolve the periapical lesion. Therefore, antimicrobial agents are incorporated into sealers to enhance their antibacterial efficacy [32].

The effectiveness of AgNPs against *E. faecalis* has already been reported for disinfection as an irrigant or intracanal medication [33]. *E. faecalis* is a bacterial strain that can be isolated from endodontically treated root canals and is the most important cause of failure [34]. Previous research reported the antimicrobial effectiveness upon incorporation of AgNPs into an MTA against *E. faecalis* [35]. Similarly, Jonaidi-Jafari et al. evaluated the antibacterial effect of different concentrations of AgNPs in Pro-Root MTA and Ca enriched cement (CEM) against different strains, including *E. faecalis.* The results showed that the addition of low percentages of AgNPs to MTA and CEM can be an alternative to increase the antimicrobial effects [36]. The incorporation of 0.15 wt% silver nanoparticles and DMAHDM (dimethylaminohexadecyl methacrylate) increased the antimicrobial activity of the sealer with no adverse effects on the sealer’s other properties [7]. According to the results of previous studies, it is possible to consider the antibacterial effects of different nanoparticles, including silver nanoparticles, added to endodontic sealers without affecting the physical properties [37,38].

The push-out bond test (POBT) determines the degree of resistance to dislodgment of a filling material when applied to the dentin of the root canal; this test shows a better assessment of bond strength than conventional shear tests [39]. The bond strength is an important property, as it minimizes the risk of the filling material dislodging during condensation forces, which can lead to reinfection and failure [40]. Despite the limitations in the variability of dental anatomy, for POBT, it was decided to use human teeth, because they are the substrate that best reproduces the clinical conditions in laboratory tests as the dentine microstructure, as no material can properly reproduce the physicochemical and morphological features as the tooth [41]. The use of natural teeth is also important to test hydraulic materials like bioceramic sealers that depend on specific environments to set [42].

Although the adhesion of the tricalcium silicate cement sealers and their ability to attach to dentinal walls may be augmented by means of bioactivity, the precipitation of hydroxyapatite crystals and the sealer fills in voids and creates frictional (mechanical) adhesion at the dentinal wall [43].

The six groups under discussion exhibited different bond strengths; however, there were no significant differences when AgNPs were added, except for Endosequence BC Sealer™ in the middle third, which we attribute to the internal anatomy of the canals since the conicity is not constant and the root canals can adopt an oval shape. Thus, we can infer that AgNPs do not affect the physical property of adhesion strength of bioceramic sealers.

An evaluation of the antibacterial and physicochemical properties of two bioceramic sealers, including compressive strength, was performed. No significant differences between BioRoot and CeraSeal^®^ were measured after 24 h in water, which coincides with us for CeraSeal^®^ at the apical third [31].

Respecting the values of bond strength, Bio-C^®^ Sealer exhibited the lowest adhesion in MPa in all thirds with or without AgNPs, which aligned with published results [44]; contrary to earlier reported results, Bio-C Sealer was the most effective adhesive to the dentin compared with other bioceramic sealers [45].

Moreover, the adhesion force of the TotalFill sealer with a single cone technique was reported. In the present study, at the middle third in CeraSeal^®^, CeraSeal^®^ + AgNPs, EndoSequence^®^ BC Sealer^TM^, and Endosequence BC Sealer™ + AgNPs, the adhesion force was higher compared to what has previously been reported [6]. Regarding the type of failure, the mixed was the most predominant, followed by the cohesive and the least prevalent adhesive, as shown in our study.

A previous study reported a maximum dislodgement force for EndoSequence^®^ BC Sealer^TM^ with and without gutta-percha obturation. However, in our study, in all thirds, the force obtained was higher for this sealer. They reported that adhesive failure was the most frequent, and we can infer that these differences are because the methodology used was different [46].

For the surface roughness test, AFM was used, since it allows obtaining images and measuring the topography of surfaces in high resolution [47]. The roughness has already been previously evaluated for dental restoration materials. It has been suggested that rough surfaces of the oral cavity accumulate between two and three times more bacteria compared to smooth surfaces; when the surface roughness of a restoration is lower, at 1 μm, a reflective surface is produced due to the wavelength of visible light [48]. The roughness of the surface influences bacterial adhesion, and the greater the roughness of the surface can cause greater bacterial accumulation. The critical roughness for increasing bacterial adhesion is 0.2 μm [49,50]. Similar work evaluated the roughness of two bioceramic sealers AH Plus Bioceramic and Well-Root ST compared to a resin-based sealer AH Plus without finding statistically significant differences between the two bioceramic sealers [51]. No statistically significant differences in roughness were observed between the groups with and without AgNPs.

The advantage of direct contact tests over the agar diffusion test is that it is independent of the diffusion properties of the tested material and media. Serial dilutions of a solution are used for MBC to determine the lowest concentration of material that would still show antibacterial properties [52].

An important limitation of this research is that it was done on planktonic bacteria. It is well established that endodontic infections are mediated by biofilms [53,54]. Despite the antibacterial efficacy of AgNPs in endodontics [55,56], further studies are required to assess the antimicrobial effect of bioceramic sealers supplemented with silver nanoparticles on endodontic biofilm.

In summary, the results obtained with the MBC test allowed us to corroborate that for both Endosequence BC Sealer™ and B Bio-C^®^ Sealer, the addition of the AgNPs improves the antimicrobial activity, since a smaller amount of sealer is required to achieve the elimination of *E. faecalis.* Therefore, the null hypothesis of the study was accepted.

## 4. Materials and Methods

### 4.1. Chemical Synthesis of NPs

For the synthesis of AgNPs, 100 mL of a 0.01 M solution of AgNO_3_ (Fermont, Monterrey, Mexico) was placed in a reaction beaker under magnetic stirring (Pc-420D, Corning, NY, USA) at room temperature, and 10 mL of deionized water containing 0.1 g of gallic acid (Sigma-Aldrich, San Luis, MO, USA) was added. The pH value of the reaction was adjusted to 11 using a 3.0 M NaOH (Sigma-Aldrich, San Luis, MO, USA) solution. The prepared AgNPs were characterized by UV-VIS spectroscopy (GENESYS 10S UV-Vis v4.006, Madison, WI, USA), dynamic light scattering (DLS) analysis (Anton Paar, Litesizer DLS 500, Graz, Austria), and transmission electron microscopy (TEM, JEOL JEM 1230, Akishima, Tokyo, Japan).

### 4.2. Push-Out Bond Strength Test

The sample calculation was made, based on parameters of the standard deviation from a previous study [6]. A probability level of α = 0.05 and a statistical power of 0.9 were considered. One hundred twenty single-root human premolars extracted for orthodontic reasons, free of caries, fissures, fractures, and with a closed apex were included. Before manipulation of samples, the organic tissue was removed by Gracey curettes (Hu-Friedy, Chicago, IL, USA) and the inorganic tissue by treatment with an ultrasonic bath (Biosonic UC-50, Coltene/Whaledent, Inc., Altstätten, Switzerland) 17% EDTA (MD-Cleanser, Meta Biomed Co., Cheongju, Republic of Korea) for 4 min and 5.25% NaOCI (Viarzoni-T, Viarden, Mexico City, Mexico) for 4 min [57] and, subsequently, sterilized by autoclave at temperature of 121 °C and 15 psi for 20 min, then kept in a sterile saline solution until use. The samples were randomly distributed using the random.org program (http://www.random.org, accessed on 9 January 2023) into six groups (*n* = 20): G1 CeraSeal^®^, G2 CeraSeal^®^ + AgNPs, G3 EndoSequence^®^ BC SealerTM, G4 EndoSequence^®^ BC SealerTM + AgNPs, G5 Bio-C^®^ Sealer, and G6 Bio-C^®^ Sealer + AgNPs. The composition of the sealers is described in Table 5.

All treatment procedures were performed by a single practitioner experienced in endodontics. The root lengths were adjusted to 15 mm [58] with a diamond disk mounted on a slow-speed handpiece with a coolant, and the working length was established at 1 mm from the apical foramen and confirmed with digital radiography. The roots were performed with K files #10 and #15 (Dentsply Sirona, Charlotte, NC, USA), followed with Protaper Next system (Dentsply Sirona, Charlotte, NC, USA) up X3 file, alternating with manual irrigation of 2 mL of 5.25% NaOCl (Viarzoni-T, Viarden, Mexico City, Mexico) between each instrument with Endo-Eze needle 27G gauge (Ultradent Products, Inc., South Jordan, UT, USA). The final irrigation was performed with 2 mL of 5.25% NaOCl (Viarzoni-T, Viarden, Mexico City, Mexico) and 5 mL of 17% EDTA (MD-Cleanser, Meta Biomed Co., Cheongju, Republic of Korea) for 1 min [59,60]. The canals were subsequently dried with sterile #30 paper points (Meta Biomed Co., Ltd., Cheongju, Republic of Korea).

The filling was performed with the single cone technique X3 and standardized gutta-percha from the Protaper Next system (Dentsply Sirona, Charlotte, NC, USA) was used. For the groups with AgNPs, for each mL of bioceramic sealer, 10 μL of AgNPs at 1070 ppm were added with a micropipette. The mixing was done manually; the final concentration of nanoparticles in the groups with sealant was 535 ppm, determined in the pilot test. An X-ray was taken of the master cone; subsequently, each of the different sealers was brought to the canal using the dispensing tip, and the master cone was embedded in the sealer until the previously established working length was reached. Excess of gutta-percha was cut with a hot medium plugger (Meta Biomed Co., Ltd., Cheongju, Republic of Korea) and vertical compaction was performed with a 50/90 hand plugger (B&L Biotech, Inc., Fairfax, VA, USA).

The samples were maintained under incubation at 36 ± 1 °C and 100% humidity for 48 h. Subsequently, three sections of 2 mm ± 0.1 thickness were obtained from each sample, which corresponded to the cervical, middle, and apical third. For the push-out bond strength test, each section was placed on the metal slab of the Universal Testing Machine (Multitest 1-d/Mecmesin, West Sussex, UK), which contained a central slot to allow free movement of endodontic plugger taper 0.06 for cervical third, taper 0.04 for middle third and taper 0.02 for apical third (Meta Biomed Co., Ltd., Cheongju, Republic of Korea) adapted to a metallic attachment. The vertical load was applied in an apical-coronal direction at a constant speed of 1 mm/min. To express the bond strength in megapascals (MPa), the recorded load in Newton (N) was divided by the area of the canal wall in mm^2^.

Bond Strength (MPa) = Dislodgement Force (N)/Adhered Surface Area (mm^2^). Where, bonded interface area (mm^2^) equals 2πrh (π = 3.1416, r = root canal radius, and h = dentin slice thickness in mm) [40].

Immediately after, each section was examined under a stereoscopic microscope (Luxeo 4Z Labomed Inc., Los Angeles, CA, USA) at 35X magnification to determine the type of failure. The types of failure were defined as cohesive, material present in the entire canal wall; adhesive, no material remains on the canal wall; and mixed, material in patches on the canal wall.

### 4.3. Surface Roughness Measurement

For all sealer groups with and without AgNPs, 1 mL was placed on a microscope slide, and incubated at 36 ± 1 °C and 100% humidity for 48 h before testing. Measurements for surface roughness were obtained from three different points in the samples using atomic force microscopy (AFM) (Nanosurf Easy Scan 2, SPM Electronics, Liestal, Switzerland) in contact mode with a silicon nitride (SiN) scanning rate of 49.5 μm/s. The values used for the short cantilever were spring constant 0.1 N/m, resonant frequency 28 kHz; length 225 μm; mean width 28 μm; thickness 1 μm; tip height 14 μm and radius <10 nm. A calibration grid of silicon oxide on a silicon substrate (Nanosurf AG, CH-4410, SPM Electronics, Liestal, Switzerland) with XY periodicity of 10 μm and a Z height of 119 nm was used to calibrate the instrument before the evaluation. The Nanosurf Easy Scan 2 software (version 1.6) was used to measure the AFM parameters. The surface roughness was quantified using roughness average (Sa), which represents the arithmetical mean of the absolute values of the scanned surface profile, and peak–valley height (Sy), which represents the absolute average value of the heights of the five highest-profile peaks and the depths of five deepest profile valleys within the sampling length on the scanned surface.

### 4.4. Disk Diffusion Test

Mueller-Hinton (M-H) agar Petri dishes were used, and the tested strain was *Enterococcus faecalis* (ATCC 29212). The inoculum was prepared on the McFarland scale (0.05) at a concentration of 10^8^ CFU/mL. The surface of each plate was swabbed in three directions with the inoculum, followed by the addition of filter paper discs impregnated with each group of fresh sealers. After 24 h of incubation at 36 ± 1 °C, the diameter of the zone of growth inhibition was measured in mm.

### 4.5. Plate Microdilution Method

The method was performed according to the Clinical and Laboratory Standards Institute (CLSI), which determines the Minimum Bactericidal Concentration (MBC) [61]. The same *E. faecalis* at 10^8^ CFU/mL was used. For groups without AgNPs, 100 mg of sealer was diluted in 0.5 mL of NaCl, whereas for groups with AgNPs, 100 mg of sealer was diluted in 0.5 mL of NaCl supplemented with 0.5 μL of AgNPs at a concentration of 535 ppm. The groups were diluted 2–128 times with 100 µL of M-H broth inoculating the bacterium at a concentration of 10^5^ CFU/mL. The MBC value was determined by visual assessment, corresponding to the well with the minimum AgNPs concentration that prevented bacterial growth after 24 h of incubation at 36 ± 1 °C.

### 4.6. Statistical Analysis

Statistical analyses were performed with the JMP program, version 10.0 (SAS Institute, Cary, NC, USA). The Shapiro–Wilk test was used to determine distribution. The differences were analyzed with the non-parametric Kruskal–Wallis test between groups and the differences between groups for thirds with Mann–Whitney U Test for push-out bond strength test, the results of roughness average with a paired student’s *t*-test, and with Mann–Whitney U Test for disk diffusion and microdilution method. Statistical significance was established with a *p*-value < 0.05.

## 5. Conclusions

Within the limitations of this in vitro study, the incorporation of AgNPs into commercial bioceramic sealers is promising for improving antimicrobial activity against resistant endodontic pathogens like *E. faecalis* without affecting their mechanical properties, such bonding strength to dentin and surface roughness, highlighting that stable materials were presented after the incorporation of the nanoparticles with improved antibacterial properties.

## Figures and Tables

**Figure 1 molecules-29-04422-f001:**
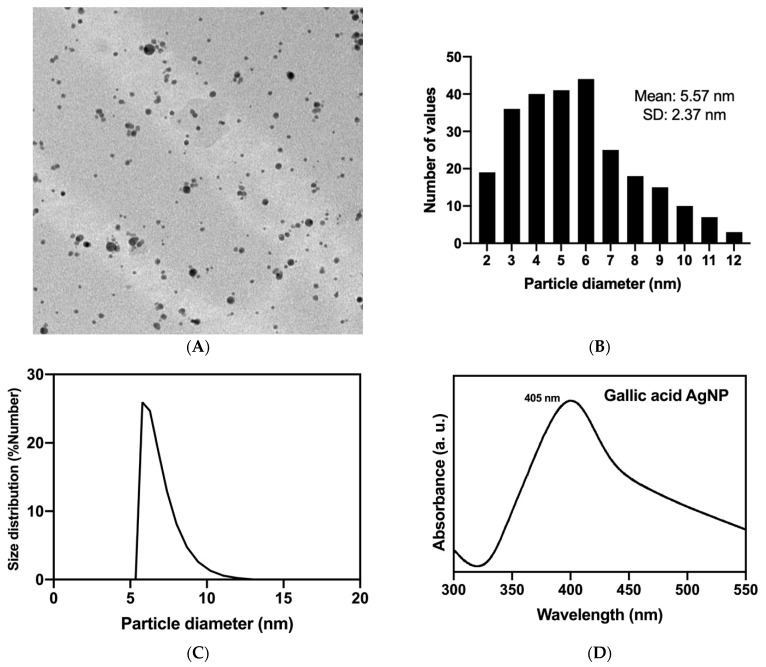
Characterization of the synthesized AgNPs. (**A**) TEM image (scale bar = 20 nm); (**B**) particle size distribution obtained from TEM image (*n* = 175); (**C**) hydrodynamic diameter; and (**D**) surface plasmon resonance spectra.

**Figure 2 molecules-29-04422-f002:**
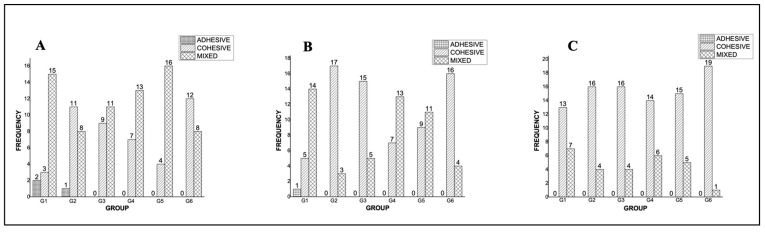
The frequency of type of failure in different portions of the sample. (**A**) Cervical third, (**B**) middle, and (**C**) apical.

**Figure 3 molecules-29-04422-f003:**
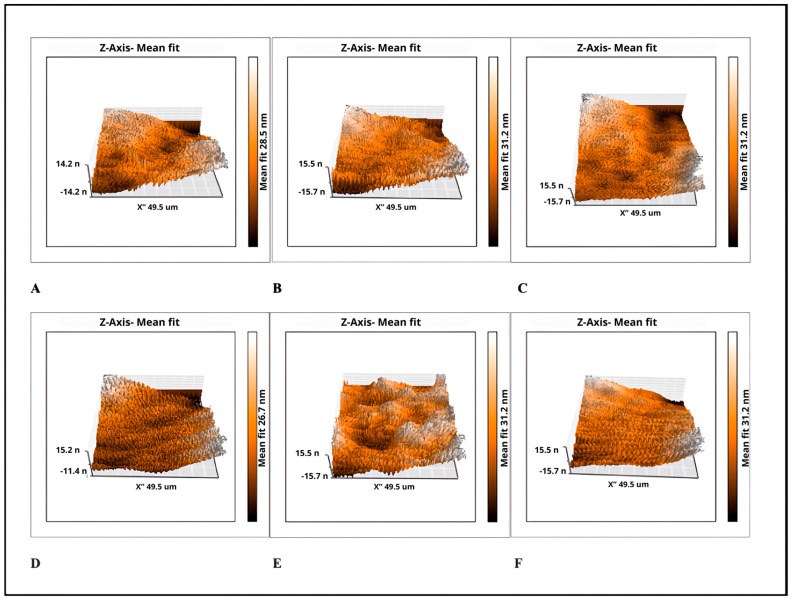
3D atomic force microscopy images. (**A**) G1 CeraSeal^®^; (**B**) G2 CeraSeal^®^ + AgNPs; (**C**) G3 EndoSequence^®^ BC Sealer^TM^; (**D**) G4 EndoSequence^®^ BC Sealer^TM^ + AgNPs; (**E**) G5 Bio-C^®^ Sealer; and (**F**) G6 Bio-C^®^ Sealer + AgNPs.

**Table 1 molecules-29-04422-t001:** Adhesion strength in MPa reported in all thirds.

Group	Cervical			Middle			Apical		
	**Mean ± SD (MPa)**	**Std. Error**	***p*-Value**	**Mean ± SD (MPa)**	**Std. Error**	***p*-Value**	**Mean ± SD** **(MPa)**	**Std. Error**	***p*-Value**
G1 CeraSeal	8.5 ± 2.8	0.6		14.9 ± 4.7	1.0		45.3 ± 13.8	3.0	
G2 CeraSeal + AgNPs	6.6 ± 3.6	0.8	0.128	14.1 ± 5.4	1.2	0.983	49.2 ± 14.2	3.1	0.96
G3 EndoSeq	4.7 ± 2.2	0.5		11.3 ± 3.6	0.8		40.4 ± 13.7	3.0	
G4EndoSeq	3.6 ± 1.7	0.3	0.648	6.7 ± 2.8	0.6	0.002	34.3 ± 16.6	3.7	0.772
+AgNPs									
G5 Bio-C Sealer	2.3 ± 0.9	0.2		4.5 ± 2.4	0.5		13.7 ± 7.6	1.7	
G6 Bio-C Sealer + AgNPs	2.6 ± 1.5	0.3	0.997	4.2 ± 2.0	0.4	0.998	14.2 ± 6.7	1.5	0.998

**Table 2 molecules-29-04422-t002:** Surface roughness values Sa and Sy of the sealers tested with and without AgNPs.

Group	Sa			Sy		
	**Mean ± SD (nm)**	**Std. Error**	***p*-Value**	**Mean ± SD (nm)**	**Std. Error**	***p*-Value**
G1 CeraSeal	4.1 ± 0.8	0.8		38.9 ± 5.6	3.2	
G2 CeraSeal + AgNPs	3.4 ± 1.0	0.5	0.723	36.1 ± 4.4	2.5	0.599
G3 EndoSeq	4.8 ± 0.2	0.1		39.2 ± 1.9	1.1	
G4 EndoSeq + AgNPs	4.6 ± 0.7	0.4	0.118	38.2 ± 2.6	1.5	0.556
G5 Bio-C Sealer	4.2 ± 0.3	0.1		40.9 ± 4.1	2.4	
G6 Bio-C Sealer + AgNPs	4.5 ± 0.4	0.2	0.438	43.9 ± 8.7	5.0	0.332

**Table 3 molecules-29-04422-t003:** Bacterial inhibition halos of groups against *E. faecalis*.

Group	Mean ± SD	Std. Error	*p*-Value
G1 CeraSeal	17.7 ± 0.8	0.4	
G2 CeraSeal + AgNPs	16.5 ± 1.6	0.9	0.275
G3 EndoSeq	16.3 ± 1.4	0.8	
G4 EndoSeq + AgNPs	20.9 ± 3.6	2.1	0.127
G5 Bio-C Sealer	15.3 ± 2.1	1.1	
G6 Bio-C Sealer + AgNPs	17.0 ± 2.8	1.6	0.376
AgNPs (535 μg/mL) †	14.2 ± 1.3	0.7	

All values are expressed in millimeters. † Control group.

**Table 4 molecules-29-04422-t004:** MBC from sealers against *E. faecalis*.

Group	Mean ± SD (μg/mL)	Std. Error	*p*-Value
G1 CeraSeal	133.7 ± 0.0	0.0	
G2 CeraSeal + AgNPs	133.7 ± 0.0	0.0	1.000
G3 EndoSeq	267.5 ± 0.0	0.0	
G4 EndoSeq + AgNPs	8.3 ± 0.0	0.0	0.025
G5 Bio-C Sealer	535.0 ± 0.0	0.0	
G6 Bio-C Sealer + AgNPs	267.5 ± 0.0	0.0	0.025
AgNPs (535 μg/mL) †	267.5 ± 0.0	0.0	

† Control group.

**Table 5 molecules-29-04422-t005:** Composition of endodontic sealers.

Name	Manufacturer	Composition *
CeraSeal^®^	Meta Biomed Co., Ltd., Cheongju, Republic of Korea	Zirconium dioxide, tricalcium silicate, dicalcium silicate, tricalcium aluminate, and thickening agent
EndoSequence^®^ BC Sealer^TM^	Brasseler USA, Savannah, GA, USA (EE. UU.)	Zirconium oxide, calcium silicates, calcium phosphate monobasic, calcium hydroxide, filler, and thickening agents
Bio-C^®^ Sealer	Angelus, Londrína, PR, Brazil	Calcium silicates, calcium aluminate, calcium oxide, zirconium oxide, iron oxide, silicon dioxide, and dispersing agent

* The compositions are based on the available manufacturer information.

## Data Availability

Data are contained within the article.

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
