# Peer review of "The Push-Out Bond Strength, Surface Roughness, and Antimicrobial Properties of Endodontic Bioceramic Sealers Supplemented with Silver Nanoparticles"

_molecules, 2024, doi:10.3390/molecules29184422_

Round 1

Reviewer 1 Report (Previous Reviewer 1)

Comments and Suggestions for Authors

                The manuscript ID: Molecules -  3181974 entitled “The Push-out Bond Strength, Surface Roughness, and Antimicrobial Properties of Endodontic Bioceramic Sealers Supplemented with Silver Nanoparticles” has been submitted for publication in Molecules - MDPI. This manuscript is focused on the development of bioceramic sealers supplemented with silver nanoparticles (AgNPs), assessing the effect of latter content on the physical-mechanical properties and antimicrobial activity of these materials.

                                    I think this work has been improved. It should be accepted for publication after modifying the following item:

-Page 10. Line 401-405. The following paragraph must be removed since it proposes aims that are far from those proposed in this study.

…..“ however, it will need more studies for their clinical applications with the purpose of evaluating whether the addition of silver nanoparticles to bioceramic sealers interferes with other physical-mechanical properties such as radiopacity, fluidity, working time, solubility, pH; as well as biocompatibility, cytotoxicity and bioactivity.”

Author Response

I think this work has been improved. It should be accepted for publication after modifying the following item:

Q1. Page 10. Line 401-405. The following paragraph must be removed since it proposes aims that are far from those proposed in this study.

…..“ however, it will need more studies for their clinical applications with the purpose of evaluating whether the addition of silver nanoparticles to bioceramic sealers interferes with other physical-mechanical properties such as radiopacity, fluidity, working time, solubility, pH; as well as biocompatibility, cytotoxicity and bioactivity.”

A1. Thank you for your suggestion, the paragraph was removed for the manuscript.

Reviewer 2 Report (New Reviewer)

Comments and Suggestions for Authors

Dear Authors,

the work is well structured and of current interest. Although to improve the scientific impact of this study, some changes are needed that I report below.

Introduction:

1. I suggest improving this section by expanding the content. In fact, I would better describe the characteristics of bioceramics and which are the most important to understand which of these can influence the effectiveness of the tested materials.

2. I would highlight the importance of having used different techniques to obtain the results, why is it necessary? Also why did you only consider E. feacalis? Is there a reason?

3. I recommend inserting the null hypothesis and refuting it in discussion.

Mat and Met:

4. I suggest modifying the structure of the entire work by ordering the sections according to the guidelines of the journal.

5. Insert the sample size in the statistics paragraph, reporting the result of the previous study.

6. How did you mix the Ag particles with the Bioceramics? Are there Bioceramics on the market that already contain these particles?

7. Insert ethics committee or etichal approval considering biological samples.

8. Suggestion to insert reference on the preparation-irrigation-closure protocol (endodontic treatment) (see https://doi.org/10.3390/app13010149 ; https://doi.org/10.3389/froh.2022.838043.)

Discussion and conclusion:

9. Well structured and formulated, I suggest to conclude with the salient points of the work to increase the impact of the work.

Comments on the Quality of English Language

Minor editing of English language required.

Author Response

Introduction

COMMENT 1

I suggest improving this section by expanding the content. In fact, I would better describe the characteristics of bioceramics and which are the most important to understand which of these can influence the effectiveness of the tested materials.

A1. In accordance with their recommendations, in the introduction section the content about bioceramics was expanded on lines 61 to 72 on page 2.

COMMENT 2 

I would highlight the importance of having used different techniques to obtain the results, why is it necessary? Also why did you only consider E. feacalis? Is there a reason?

A2. On lines 93 to 99 we explain the reasons why it was decided to perform microbiological tests only on the E. faecalis.

COMMENT 3

I recommend inserting the null hypothesis and refuting it in discussion.

A3. The null hypothesis was added on lines 104 to106 in introduction and accepted in discussion on line 248.

Mat and Met:

COMMENT 4

I suggest modifying the structure of the entire work by ordering the sections according to the guidelines of the journal.

A4. To structure the manuscript, the instructions for the authors on the official page https://www.mdpi.com/journal/molecules/instructions were revised, as well as the use of the journal template.

COMMENT 5

Insert the sample size in the statistics paragraph, reporting the result of the previous study.

A5. The previous study doi: 10.1016/j.jdent.2019.07.007, it was only used as a basis to calculate the sample size using a formula for estimating the population mean and the data used was the standard deviation. In this study the n was 10 per subgroup, in our study it was 20 per group, to avoid confusion is not mentioned in the document. We described on lines 261 and 262.

COMMENT 6

How did you mix the Ag particles with the Bioceramics? Are there Bioceramics on the market that already contain these particles?

A6. In lines 291 to 294 we explain how mixed the bioceramic and AgNps sealers. There are still no commercial preparations with these characteristics on the market.

COMMENT 7

Insert ethics committee or ethical approval considering biological samples.

A7. The ethics committee approval is found in the Institutional Review Board Statement section on page 11.

COMMENT 8

Suggestion to insert reference on the preparation-irrigation-closure protocol (endodontic treatment) (see https://doi.org/10.3390/app13010149 ; https://doi.org/10.3389/froh.2022.838043.)

A8. Thanks for the suggestion, the references were reviewed and included on line 287.

Discussion and conclusion:

COMMENT  9

Well structured and formulated, I suggest to conclude with the salient points of the work to increase the impact of the work.

A9. We accepted your recommendation and described the importance points derived from the study on lines 360 to 365.

Round 2

Reviewer 2 Report (New Reviewer)

Comments and Suggestions for Authors

Dear Authors,

I consider the correction of the text to be satisfactory. The work is now suitable for publication. I recommend revising the text for English Language before finishing. Well done.

Comments on the Quality of English Language

I recommend reviewing the entire text for correction of the quality of English language, especially the conclusions.

This manuscript is a resubmission of an earlier submission. The following is a list of the peer review reports and author responses from that submission.

Round 1

Reviewer 1 Report

Comments and Suggestions for Authors

Comments for authors:

                                           The manuscript ID: Molecules - 3115591 entitled “The Push-out Bond Strength, Surface Roughness, and Antimicrobial Properties of Endodontic Bioceramic Sealers Supplemented with Silver Nanoparticles” has been submitted for publication in Molecules - MDPI. This manuscript is focused on the development of bioceramic sealers supplemented with silver nanoparticles (AgNPs), assessing the effect of latter content on the physical-mechanical properties and antimicrobial activity of these materials.

                                    I think this work has enough scientific relevance. however, it needs major revision. It should not be published before analyzing some topics that must be considered carefully.

-Page 2. Line 22. I think the null hypothesis is wrong. Please, just state the study aim of the work in a clear way.

- Page 3. Lines 104-105. The author stresses that there are no significant differences in the cervical value for G1 and G2 (Table 1). Please, it should be reviewed.

- Page 6. Line 163. ………..The incorporation of 0.15% silver nanoparticles……..

This percentage should be specified. Maybe, is it wt. %?

- This sentence (……in the present study ….) is repeated a lot in the text. It should be modified.

-Page 7. Lines 222-230. This paragraph is confusing. It distracts the reader. It should be removed.

-Page 8. Line 243. “Therefore, the null hypothesis of the study was accepted”. I think this sentence must be removed. This contributes to a confusing reading.

Reviewer 2 Report

Comments and Suggestions for Authors

In this paper, Authors propose the characterization of endodontic bioceramic sealers, modified with AgNPs prepared through chemical reduction. This study presents many limitations, due to the very low degree of novelty of the proposed research. Neither the synthesis, nor the preparation and characterization of the material are new, as demonstrated by many similar papers on the topic published in the last years: 10.4103/JCD.JCD_266_17, 10.5395/rde.2021.46.e38, 10.1177/2280800019851771, 10.1007/s10266-020-00507-x, 10.1016/j.joen.2023.07.011, 10.21608/jfcr.2022.245560 (just to cite a few). A detailed list of comments is reported below:

-       It is unproper (from a chemical point of view) to define bioceramics as “non-metallic” materials. In fact, many of them contain elements like Zirconium, Calcium, etc., which are indeed classified as metals in the periodic table. Please revise bioceramic definition in the introduction section.

-       The antibacterial activity of Ca(OH)2 is known, but the discussion about this point should be supported by references.

-       All the “Results” section should be deeply revised. ALL experimental data are presented in an unproper way, from the statistical point of view. Mean values and standard deviations must be listed with the correct number of significant digits. As an example, for sample “G1 CeraSeal”, 8.52 ± 2.82 MPa should be 8.5 ± 2.8 MPa or, better, 8 ± 3 MPa. All experimental data must be accompanied by corresponding errors.

-       Particles size distribution histogram should be revised. Bin size should be adjusted to reveal either a Gaussian or a LogNormal distribution. The histogram bars should be fitted by one of these two distributions and the corresponding mean size and standard deviation should be retrieved again.

-       Line 105: Do you mean “Table 1”?

-       Figure 3: why is “panel A” labelled as “line fit” and all the others as “mean fit”?

-       How did you choose the concentration of AgNPs for control experiments?

-       The AgNPs used in this work are ultrafine (much below 50 nm in diameter). Hence, there is a significant nano-toxicological risk associated with their use. Moreover, ingestion is one of the most dangerous routes for NPs entry in the human body. This aspect is not even cited in the manuscript. Neither the absence of NP release nor the cytotoxicity of these AgNPs was tested.

-       Authors claim that their composites containing AgNPs are antimicrobial, but the results from Agar diffusion tests are not encouraging. When in a composite, the antimicrobial action of AgNPs is mainly exerted trough the release of Ag+ ions and the catalytic production of reactive oxygen species. Probably these mechanisms do not function in the bioceramic/AgNPs composite. Explanation to the results (lines 222-234) does not consider this, and reference 51 (used to support this explanation) is not pertinent (AgNPs not in a composite, another microorganism, etc.).

-       Authors should describe how the AgNP-bioceramic composites were prepared. Which is the final concentration of AgNPs in the composite?

-       Correct bacterial concentration at line331 (superscript missing).

-       Conclusions are too short. Please better describe the future perspectives of this study.

Comments on the Quality of English Language

English quality is good. Some minor refinements anre necessary (typos).